# PREDICTING MULTIPLE ACTIONS FOR STOCHASTIC CONTINUOUS CONTROL

## ABSTRACT

We introduce a new approach to estimate continuous actions using actor-critic algorithms for reinforcement learning problems. Policy gradient methods usually predict one continuous action estimate or parameters of a presumed distribution (most commonly Gaussian) for any given state which might not be optimal as it may not capture the complete description of the target distribution. Our approach instead predicts $M$ actions with the policy network (actor) and then uniformly sample one action during training as well as testing at each state. This allows the agent to learn a simple stochastic policy that has an easy to compute expected return. In all experiments, this facilitates better exploration of the state space during training and converges to a better policy.

## 1 INTRODUCTION

Reinforcement learning is a traditional branch of machine learning which focuses on learning complex tasks by assigning rewards to agents that interact with their environment. It has recently gained momentum thanks to the combination of novel algorithms for continuous control with deep learning models, sometimes even matching human performance in tasks such as playing video games and manipulating objects Mnih et al. (2015); Silver et al. (2016). Recent methods for continuous control problems like Deep Deterministic Policy Gradient (DDPG) Lillicrap et al. (2016), Asynchronous Advantage Actor Critic (A3C) Mnih et al. (2016) use actor-critic architectures, where an action function is learned by mapping states to actions. DDPG works well on many tasks, but it does not model the uncertainty in actions as it produces a point estimate of the action distribution over states. The actor is forced to deterministically choose an action for every state. A3C and other stochastic policy gradient algorithms output distribution parameters (e.g. Gaussian distributions) instead of point estimate, which can be sampled for action values.

As a simple example where this is sub-optimal, consider the inverted pendulum task, where a pendulum is attached to a cart and the agent needs to control the one dimensional movement of the cart to balance the pendulum upside down. A deterministic agent chooses a single action for every state. This breaks the inherent symmetry of the task. When the cart in not moving and the pendulum is hanging down, two actions are equally promising: either moving left or right. The distribution parameter estimation (e.g. A3C) might work better in this case as there are only two good options, but in cases when there are more than two good actions to select, this will not be optimal. In our approach we allow the agent to suggest multiple actions, which enables it to resolve cases like this easily.

Further, we observe that a deterministic behavior of DDPG can lead to sub-optimal convergence during training. The main limitation is that, especially in the beginning of the learning procedure, the actor favors actions that lead to a good immediate reward but might end up being far from the globally optimal choice.

This work is based on the intuition that if the actor is allowed to suggest, at each time step, multiple actions rather than a single one, this can render the resulting policy non-deterministic, leading to a better exploration of the entire solution space as well as a final solution of potentially higher quality. This can also eliminate the external exploration mechanisms required during training e.g. Ornstein-Uhlenbeck process noise Uhlenbeck & Ornstein (1930), parameter noise Plappert et al. (2017) or differential entropy of normal distribution.

Here, we introduce an algorithm, which we refer to as Multiple Action Policy Gradients (MAPG), that models a stochastic policy with several point estimates and allows to predict a pre-defined number $M$ of actions at each time step, extending any policy gradient algorithm with little overhead. We will demonstrate the working of this algorithm by adapting DDPG Lillicrap et al. (2016) to use MAPG.

Another benefit of the proposed method is that the variance of the predicted actions can give additional insights into the decision process during runtime. A low variance usually implies that the model only sees one way to act in a certain situation. A wider or even multi-modal distribution suggests that there exist several possibilities given the current state.

We evaluate the proposed method on six continuous control problems of the OpenAI Gym Brockman et al. (2016) as well as a *deep driving* scenario using the TORCS car simulator Wymann et al. (2014). For a fair evaluation we directly compare DDPG to our MAPG without changing hyper-parameters or modifying the training scheme. In all experiments, we show an improved performance using MAPG over DDPG. To verify if MAPG helps in better exploration during training, we also analyze MAPG under no external exploration policy.

## 2 RELATED WORK

There is currently a wide adoption of deep neural networks for reinforcement learning. Deep Q Networks (DQN) Mnih et al. (2015) directly learn the action-value function with a deep neural network. Although this method can handle very high dimensional inputs, such as images, it can only deal well with discrete and low dimensional action spaces. Guided Policy Search Levine & Koltun (2013) can exploit high and low dimensional state descriptions by concatenating the low dimensional state to a fully connected layer inside the network.

Recent methods for continuous control problems come in two flavours, *vanilla policy gradient methods* which directly optimize the policy and *actor-critic methods* which also approximate state-value function in addition to policy optimization. Trust Region Policy Optimization (TRPO) Schulman et al. (2015) and Proximal Policy Optimization Algorithms Schulman et al. (2017) can be used as vanilla policy gradient as well as actor-critic methods. Whereas, Deep Deterministic Policy Gradient (DDPG) Lillicrap et al. (2016) and Asynchronous Advantage Actor Critic (A3C) Mnih et al. (2016) use actor-critic architectures, where state-action function is learned to calculate policy gradients.

Stochastic Value Gradients (SVG) Heess et al. (2015), Generalized Advantage Estimation (GAE) Schulman et al. (2015), A3C, TRPO all use *stochastic policy gradients* and predict action probability distribution parameters. The action values are then sampled from the predicted distribution. A parametrized normal distribution is most commonly used as action distribution. This means that this formulation models a kind of action noise instead of the true action distribution. For example a distribution with two modes cannot be modeled with a Gaussian.

DDPG Lillicrap et al. (2016) which extends DPG Silver et al. (2014) uses *deterministic policy gradients* and achieves stability when using neural networks to learn the actor-critic functions. The limitation of DDPG is that it always gives a points which may not be desired in stochastic action problems.

Lazaric et al. (2007) estimate stochastic action values using a sequential Monte Carlo method (SMC). SMC has actor and critic models where the actor is represented by Monte Carlo sampling weights instead of a general function approximator like a neural network. SMC learning works well in small state space problems, but cannot be extended directly to high dimensional non-linear action space problems.

Similar to our idea of predicting multiple instead of one output, but originating from the domain of supervised learning, is Multiple Hypothesis Prediction Rupprecht et al. (2017), which in turn is closely related to Multiple Choice Learning Lee et al. (2016) and Lee et al. (2017). In this line of work, the model is trained to predict multiple possible answers for the given task. Specific care has to be taken since often in supervised datasets not all possible outcomes are labeled, this leading to loss functions that contain an $\arg\min$-like term and, as such, are hard to differentiate.

## 3 THE MULTIPLE ACTION POLICY GRADIENT ALGORITHM

In this section we will describe in detail how multiple action policy gradients can be derived and compare it to DDPG. We will then analyze the differences to understand the performance gain.

### 3.1 BACKGROUND

We investigate a typical reinforcement learning setup Sutton & Barto (1998) where an agent interacts with an environment $E$. At discrete time steps $t$, the agent observes the full state $s_t \in \mathcal{S} \subset \mathbb{R}^c$, and after taking action $a_t \in \mathcal{A} \subset \mathbb{R}^d$, it receives the reward $r_t \in \mathbb{R}$. We are interested in learning a policy $\pi : \mathcal{S} \rightarrow \mathcal{P}(\mathcal{A})$, that produces a probability distribution over actions for each state. Similarly to other algorithms, we model the environment as a Markov Decision Process (MDP) with a probabilistic transition between states $p(s_{t+1}|s_t, a_t)$ and the rewards $r(s_t, a_t)$.

We associate a state with its current and (discounted with $\gamma \in [0, 1]$) future rewards by using

$$R_t = \sum_{i=1}^{T} \gamma^{i-t} r(s_i, a_i). \tag{1}$$

Since $\pi$ and $E$ are stochastic, it is more meaningful to investigate the expected reward instead. Thus, the agent tries to find a policy that maximizes the expected discounted reward from the starting state distribution $p(s_1)$.

$$J = \mathbb{E}_{r_i, s_i \sim E, a_i \sim \pi}(R_1) \tag{2}$$

Here, it is useful to investigate the recursive Bellman equation that associates a value to a state-action pair:

$$Q^\pi(s_t, a_t) = \mathbb{E}_{r_t, s_{t+1} \sim E}[r(s_t, a_t) + \gamma \mathbb{E}_{a_{t+1} \sim \pi}[Q^\pi(s_{t+1}, a_{t+1})]] \tag{3}$$

Methods such as (D)DPG use a deterministic policy where each state is deterministically mapped to an action using a function $\mu : \mathcal{S} \rightarrow \mathcal{A}$ which simplifies Equation 3 to

$$Q^\mu(s_t, a_t) = \mathbb{E}_{r_t, s_{t+1} \sim E}[r(s_t, a_t) + \gamma Q^\mu(s_{t+1}, \mu(s_{t+1}))]. \tag{4}$$

In Q-learning Watkins & Dayan (1992), $\mu$ selects the highest value action for the current state:

$$\mu(s_t) = \arg\max_{a_t}(Q(s_t, a_t)) \tag{5}$$

The $Q$ value of an action is approximated by a critic network which estimates $Q^\mu(s_t, a_t)$ for the action chosen by the actor network.

### 3.2 ALGORITHM

The key idea behind predicting multiple actions is that it is possible to learn a stochastic policy as long as the inner expectation remains tractable. Multiple action prediction achieves this by predicting a fixed number $M$ of actions $\rho : \mathcal{S} \rightarrow \mathcal{A}^M$ and uniformly sampling from them. The expected value is then the mean over all $M$ state-action pairs. The state-action value can then be defined as

$$Q^\rho(s_t, a_t) = \mathbb{E}_{r_t, s_{t+1} \sim E}\left[r(s_t, a_t) + \gamma \frac{1}{M} \sum_{m=1}^{M} Q^\rho(s_{t+1}, \rho_m(s_{t+1}))\right]. \tag{6}$$

This is beneficial since we not only enable the agent to employ a stochastic policy when necessary, but we also approximate the action distribution of the policy with multiple samples instead of one.

There exists an intuitive proof that the outer expectation in Equation 6 will be maximal if and only if the inner $Q^\rho$ are all equal. The idea is based on the following argument: let us assume $\rho$ as an optimal policy maximizing Equation 2. Further, one of the $M$ actions $\rho_j(s_{t+1})$ for a state $s_{t+1}$ has a lower expected return than another action $k$.

$$Q^\rho(s_{t+1}, \rho_j(s_{t+1})) < Q^\rho(s_{t+1}, \rho_k(s_{t+1})) \tag{7}$$

Then there exists a policy $\rho^*$ that would score higher than $\rho$ that is exactly the same as $rho$ exept that it predicts action $k$ instead of $j$: $\rho_j^*(s_{t+1}) := \rho_k(s_{t+1})$. However, this contradicts the assumption

---

**Algorithm 1** MAPG algorithm

---

Modify actor network $\mu(s|\theta_\mu)$ to output $M$ actions, $A_t = \{\rho_1(s_t), \ldots, \rho_M(s_t)\}$.
Randomly initialize actor $\mu(s|\theta_\mu)$ and critic $Q(s|\theta_Q)$ network weights.
Initialize target actor $\mu'$ and critic $Q'$ networks, $\theta'_\mu \leftarrow \theta_\mu$ and $\theta'_Q \leftarrow \theta_Q$.
**for** episode $= 1$ **to** $N$ **do**
   Initialize random process $\mathcal{N}$ for exploration.
   Receive initial observation/state $s_1$.
   **for** $t = 1$ **to** $T$ **do**
      Predict $M$ action proposals $A_t = \{\rho_1(s_t), \ldots, \rho_M(s_t)\} = \mu'(s_t|\theta_\mu)$.
      Uniformly sample an action $j$ from $A_t$: $a_t^j = \rho_j(s_t) + \mathcal{N}_t$.
      Execute action $a_t^j$ and observe reward $r_t$ and state $s_{t+1}$.
      Store transition $(s_t, a_t^j, r_t, s_{t+1})$ to replay buffer $R$.
      Sample a random batch of size $B$ from $R$.
      Set $y_i = r_i + Q'(s_{i+1}, \mu'(s_{i+1}|\theta'_\mu)|\theta'_Q)$.
      Update critic by minimizing the loss,
$$L = \tfrac{1}{B} \sum_i (y_i - Q(s_i, a_i^j|\theta_Q))^2$$
      Update all actor weights connected to $a_t^j$.
$$\nabla_{\theta^\mu} J \approx \tfrac{1}{B} \sum_i \nabla_{a_i^j} Q(s, a|\theta^Q)|_{s=s_i, a=a_i^j} \nabla_{\theta_\mu - \theta_\mu^{\{1 \ldots M\}} + \theta_\mu^j} \mu(s|\theta^\mu)|_{s_i}$$
      Update the target networks:
$$\theta'_\mu \leftarrow \tau\theta'_\mu + (1-\tau)\theta_\mu$$
$$\theta'_Q \leftarrow \tau\theta'_Q + (1-\tau)\theta_Q$$
   **end for**
**end for**

---

that we had learned an optimal policy beforehand. Thus in an optimal policy all $M$ action proposals will have the same expected return. More informal, this can also be seen as a derivation from the training procedure. If we always select a random action from the $M$ proposals, they should all be equally good since the actor cannot decide which action should be executed.

This result has several interesting implications. From the proof, it directly follows that it is possible - and sometimes necessary - that all proposed actions are identical. This is the case in situations where there is just one single right action to take. When the action proposals do not collapse into one, there are two possibilities: either it does not matter what action is currently performed, or all proposed actions lead to a desired outcome.

Naturally, the set of stochastic policies includes all deterministic policies, since a deterministic policy is a stochastic policy with a single action having probability density equal to one. This means that in theory we expect the multiple action version of a deterministic algorithm to perform better or equally well, since it could always learn a deterministic policy by predicting $M$ identical actions for every state.

Algorithm 1 outlines the MAPG technique. The main change is that the actor is modified to produce $M$ instead of one output. For every timestep one action $j$ is then selected. When updating the actor network, a gradient is only applied to the action (head) that was selected during sampling. Over time each head will be selected equally often, thus every head will be updated and learned during training.

## 4 EXPERIMENTS

In this section we will investigate and analyze the performance of MAPG in different aspects. First, we compare scores between DDPG, A3C and MAPG on six different tasks. Second, we analyze the influence of the number of actions on the performance by training agents with different $M$ on five tasks. Further, to understand the benefit of multiple action prediction, we observe the variance over actions of a trained agent: the goal is to analyze for which states the predicted actions greatly differ from each other and for which ones they collapse into a single choice instead. Finally, we compare the performance of DDPG and MAPG without any external noise for exploration during training.

Table 1: Tasks used for evaluation

| TASK | ACTION DIMENSION | STATE DIMENSION | DESCRIPTION |
|---|---|---|---|
| PENDULUM | 1 | 3 | PENDULUM ON A CART. |
| HOPPER | 3 | 11 | ONE LEGGED ROBOT. |
| WALKER2D | 6 | 17 | TWO DIMENSIONAL BIPEDAL ROBOT. |
| HUMANOID | 17 | 376 | THREE DIMENSIONAL BIPEDAL ROBOT. |
| HALFCHEETAH | 6 | 17 | TWO LEG ROBOT. |
| SWIMMER | 2 | 6 | THREE JOINT SWIMMING ROBOT. |
| TORCS | 3 | 29 | CONTROL CAR IN 3D SIMULATION. |

Table 2: Average score $\pm 3\sigma$ over 100 episodes for Mujoco tasks with different $M$. For better readability we denote the highest mean score for each task in bold. Corresponding boxplots can be found in Figure 1a and 1b and the appendix.

| ENVIRONMENT | DDPG | A3C | $M = 10$ | $M = 20$ | $M = 50$ |
|---|---|---|---|---|---|
| HOPPER-V1 | $603 \pm 76$ | $532 \pm 105$ | $824 \pm 94$ | $\mathbf{923} \pm 90$ | $732 \pm 34$ |
| WALKER2D-V1 | $960 \pm 72$ | $764 \pm 112$ | $1297 \pm 70$ | $1319 \pm 50$ | $\mathbf{1589} \pm 45$ |
| HUMANOID-V1 | $1091 \pm 65$ | $281 \pm 40$ | $\mathbf{1248} \pm 115$ | $1112 \pm 75$ | $1212 \pm 110$ |
| HALFCHEETAH-V1 | $4687 \pm 455$ | $3803 \pm 125$ | $\mathbf{6659} \pm 570$ | $4116 \pm 85$ | $4333 \pm 70$ |
| SWIMMER-V1 | $38 \pm 7$ | $33 \pm 10$ | $\mathbf{51} \pm 6$ | $41 \pm 4$ | $40 \pm 2$ |

## 4.1 SETUP

In all our experiments, we use five continuous control tasks from the Mujoco Simulator Todorov et al. (2012) and a driving task for The Open Racing Car Simulator (TORCS). A detailed description about the tasks is given in Table 1. We use the OpenAI Gym Brockman et al. (2016) and OpenAI baselines Hesse et al. (2017) for evaluating our experiments.

The base actor and critic networks are fixed in all experiments. Each network has two fully connected hidden layers with 64 units each. Each fully-connected layer is followed by a ReLU non-linearity. The actor network takes the current observed state $s_t$ as input and produces $M$ actions $a_t^{(m)} \in [-1, 1]^d$ by applying $\tanh$. From $M$ actions $a_t^{(m)}$, a single action $a_t$ is randomly chosen with equal probability. The critic uses the current state $s_t$ and action $a_t$ as input and outputs a scalar value ($Q$-value). In the critic network, the action value is concatenated with the output of the first layer followed by one hidden layer and an output layer with one unit.

The critic network is trained by minimizing the mean square loss between the calculated discounted reward and the computed $Q$ value. The actor network is trained by computing the policy gradient from the $Q$-value of the chosen action. The network weights of the last layer are only updated for the selected action. Ornstein-Uhlenbeck process noise is added to the action values from the actor for exploration. The training is done for a total of two million steps in all tasks.

For A3C training, we use same actor-critic networks as for earlier experiment. The output of actor network is a mean vector ($\mu_a$) (one for each action value) and a scalar standard deviation ($\sigma^2$, shared for all actions). The actions values are sampled from the normal distribution ($\mathcal{N}(\mu_a, \sigma^2)$). We used differential entropy of normal distribution to encourage exploration with weight $10^{-4}$. In our experiments, A3C performed poorly than DDPG in all tasks and was not able to learn a good policy for Humanoid task.

## 4.2 MUJOCO EXPERIMENTS

For more meaningful quantitative results, we report the average reward over 100 episodes with different values of $M$ for various tasks in 2. For all environments except HUMANOID we already score higher with $M = 5$. The lower performance in the HUMANOID task might be explained by the drastically higher dimensionality of the world state in this task which makes it more difficult to observe.

The scores of policy based reinforcement learning algorithms can vary a lot depending on network hyper-parameters, reward function and codebase/framework as outlined in Henderson et al. (2017). To minimize the variation in score due to these factors, we fixed all parameters of different algorithms and only studied changes on score by varying $M$. Our metric for performance in each task is average reward over 100 episodes by an agent trained for 2 million steps. This evaluation hinders actors with high $M$ since in every training step only a single out of the $M$ actions will be updated per state. Thus, in general actors with higher number of action proposals, will need a longer time to learn a meaningful distribution of action.

We show a plot for the scores in the HOPPER and WALKER2D environments in Figure 1a and 1b, where we can see that the overall score increases with $M$.

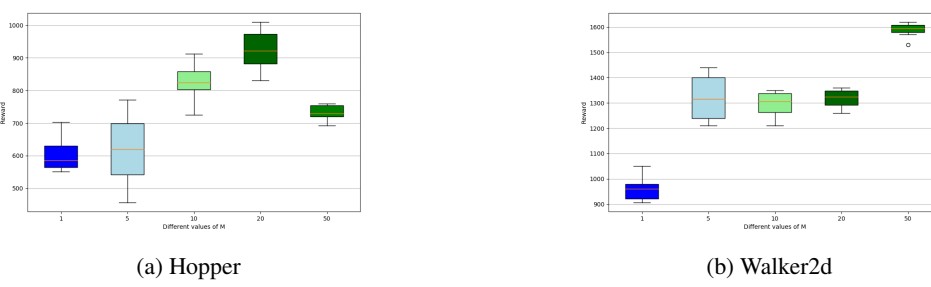

(a) Hopper                    (b) Walker2d

Figure 1: Variation in score of Hopper and Walker2d with different values of $M$.

In Figure 2, we studied the variance in action values for $M = 10$ during training together with the achieved reward. The standard deviation of actions generated by MAPG decreases with time. As the network converges to a good policy (increase in expected reward) the variation in action values is reduced. However there are some spikes in standard deviation even when network is converged to a better policy. It shows that there are situations in which the policy sees multiple good actions (with high $Q$-value) which can exploited using MAPG.

## 4.3 VARIANCE ANALYSIS

We use the simple Pendulum environment to analyze the variance during one episode. The task is the typical inverted pendulum task, where a cart has to be moved such that it balances a pendulum in an inverted position. Figure 3 plots standard deviation and the angle of the pendulum. Some interesting relationships can be observed. The variance exhibits two strong spikes that coincide with an angle of 0 degrees. This indicates that the agent has learned that there are two ways it can swing up the pole: either by swinging it clockwise or counter clockwise. A deterministic agent would need to pick one over the other instead of deciding randomly. Further, once the target inverted pose (at 180 degrees) is reached the variance does not go down to 0. This means that for the agent a slight jitter seems to be the best way to keep the pendulum from gaining momentum in one or the other direction.

With this analysis we could show that a MAPG agent can learn meaningful policies. The variance over predicted actions can give additional insight into the learned policy and results in a more diverse agent that can for example swing up the pole in two different directions instead of picking one.

## 4.4 EFFECT ON EXPLORATION

Here, we study the effect of MAPG on exploration during training. We compare the performance of DDPG and MAPG during training with and without any external noise on Pendulum and HalfCheetah environments. Figure 4 shows the average reward during training with DDPG and MAPG $M = 10$. The policy trained using MAPG converges to better average reward than DDPG in both cases. Moreover, the performance of MAPG without any external exploration is comparable to DDPG with added exploration noise. This means MAPG can explore the state space enough to find a good policy.

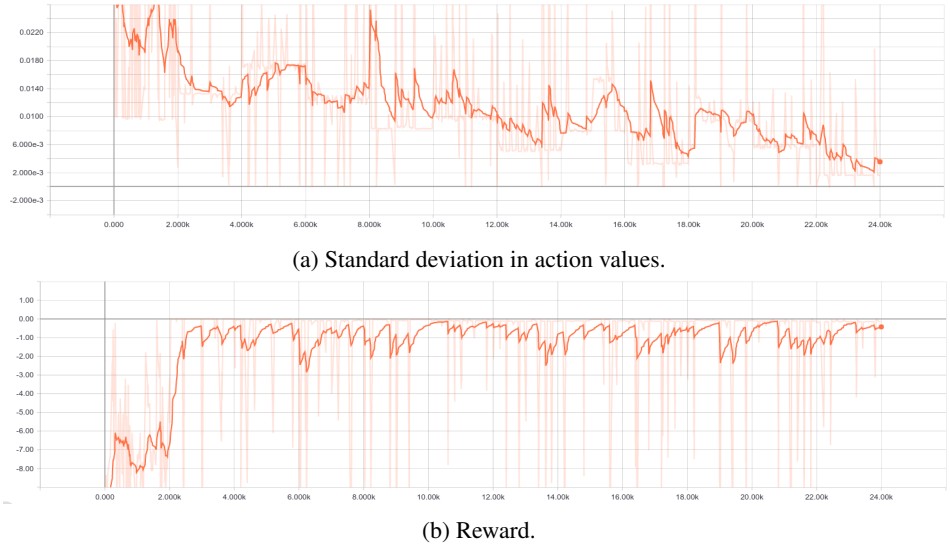

(a) Standard deviation in action values.

(b) Reward.

Figure 2: Standard deviation and reward with $M = 10$ for the Pendulum task during training.

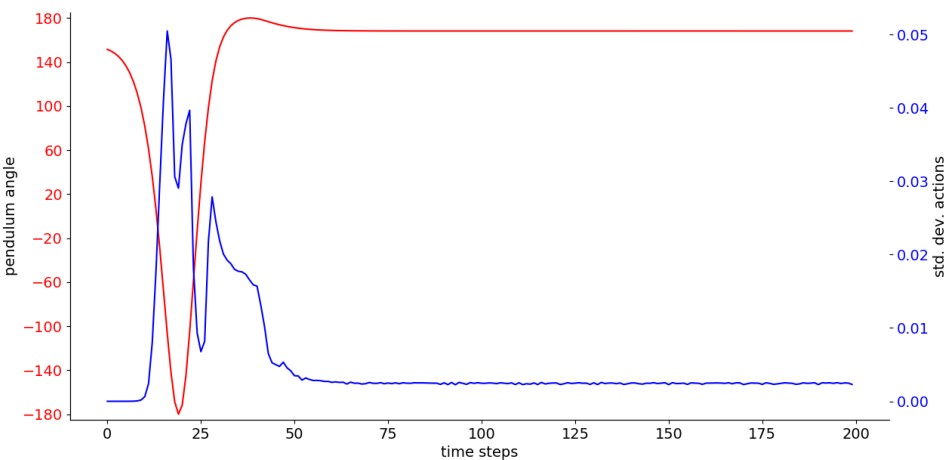

Figure 3: Standard deviation and angle during one episode of the Pendulum environment. An angle of $\pm 180$ is the target inverted pose. $0$ is hanging downwards.

In the Half Cheetah environment we can see that using exploration creates a much bigger performance difference between DDPG and MAPG than without. The difference sets in after about 500 epochs. This is an indication that in the beginning of training the actions predicted by MAPG are similar to the one from DDPG. The noise later helps to pull the $M$ actions apart such that they find individual loss minima, leading to a more diverse policy with better reward.

## 4.5 TORCS

TORCS (The Open Racing Car Simulator) is an open source 3D car racing simulator. It provides an interface for agents to drive the cars. During training, the reward was set proportional to component of car velocity along direction of road $v * \cos(\alpha)$, where $\alpha$ is the angle between the velocity vector and the center line of the track. This reward encourages forward motion. The car's sensor data (velocity, distance from road edges etc.) is used as input state and steer, brake, accelerate as actions at each time step.

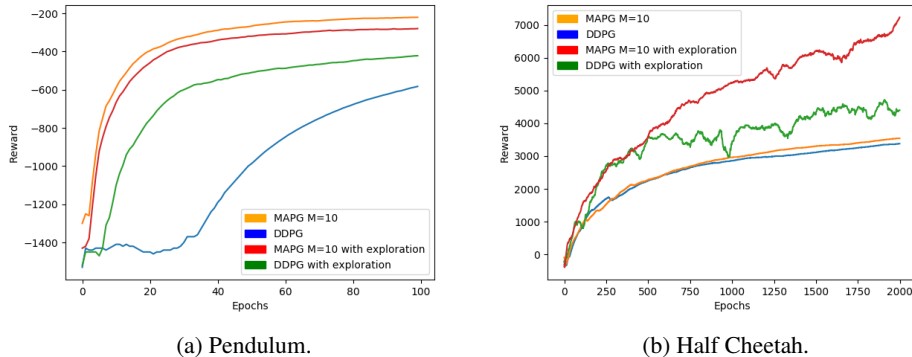

(a) Pendulum.            (b) Half Cheetah.

Figure 4: Performance curves for two environments with and without external exploration noise on DDPG and MAPG: original DDPG with OU process noise (green), DDPG without any exploration noise (blue), MAPG (M=10) with OU process noise (red) and MAPG (M=10) without OU process noise (orange).

In our experiments, MAPG with $M = 10$ was able to complete multiple laps of the track, whereas the DDPG based agent could not complete even one lap of track. The average distance traveled over 100 episodes by DDPG is 807 and 5882 (both in meters) for MAPG agent.

Similar to our other experiments we find that MAPG agents explore more possibilities due to their stochastic nature and can then learn more stable and better policies.

## 5 CONCLUSION

In this paper, we have proposed MAPG, a technique that leverages multiple action prediction to learn better policies in continuous control problems. The proposed method enables a better exploration of the state space and shows improved performance over DDPG. As indicated by exploration experiments, it can also be a used as a standalone exploration technique, although more work needs to be done in this direction. Last but not least, we conclude with interesting insights gained from the action variance. There are several interesting directions which we would like to investigate in the future. The number of actions $M$ is a hyper-parameter in our model that needs to be selected and seems to be task specific. In general, the idea of predicting multiple action proposals can be extended to other on- or off-policy algorithms, such as NAF Gu et al. (2016) or TRPO. Evaluating MA-NAF and MA-TRPO will enable studying the generality of the proposed approach.

## ACKNOWLEDGMENTS

Will be added after anonymous review.

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

APPENDIX

In the following we display the box plots similar to Figure 1a and 1b for the remaining tasks.

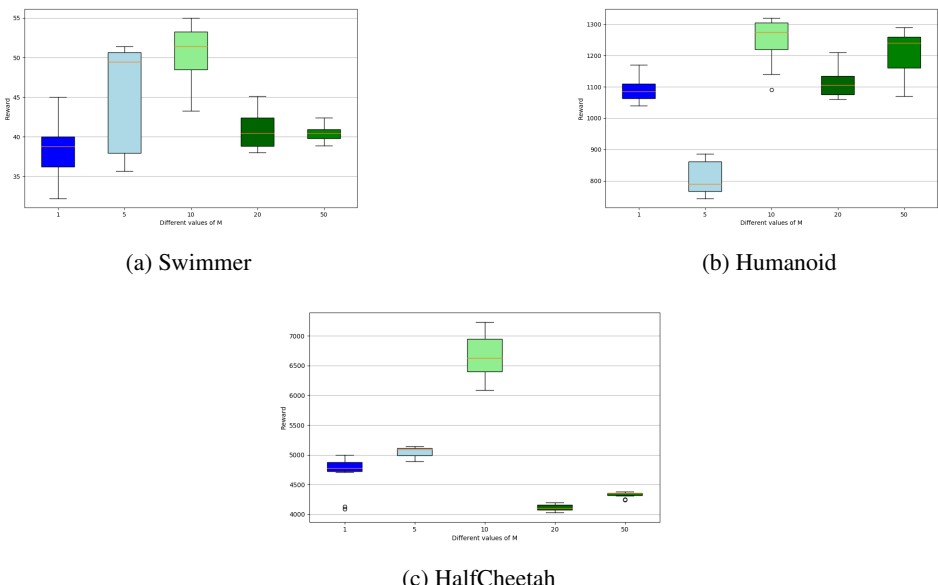

(a) Swimmer

(b) Humanoid

(c) HalfCheetah

Figure 5: Variation in score of (from top left) Swimmer, Humanoid and HalfCheetah with different values of $M$.

