# OpenReview forum: "Predicting Multiple Actions for Stochastic Continuous Control"
_ICLR.cc/2018/Conference — Reject_

### Official Review · AnonReviewer2 · 2017-11-26
**Lack of engagement with relevant prior work**

**Rating:** 3
**Confidence:** 4

**Review:**

This work introduces a particular parametrization of a stochastic policy (a uniform mixture of deterministic policies). They find this parametrization, when trained with stochastic value gradient outperforms DDPG on several OpenAI gym benchmarks.

This paper unfortunately misses many significant pieces of prior work training stochastic policies. The most relevant is [1] which should definitely be cited. The algorithm here can be seen as SVG(0) with a particular parametrization of the policy. However, numerous other works have examined stochastic policies including [2] (A3C which also used the Torcs environment) and [3].

The wide use of stochastic policies in prior work makes the introductory explanation of the potential benefits for stochastic policies distracting, instead the focus should be on the particular choice and benefits of the particular stochastic parametrization chosen here and the choice of stochastic value gradient as a training method (as opposed to many on-policy methods).

The empirical comparison is also hampered by only comparing with DDPG, there are numerous stochastic policy algorithms that have been compared on these environments. Additionally, the DDPG performance here is lower for several environments than the results reported in Henderson et al. 2017 (cited in the paper, table 2 here, table 3 Henderson) which should be explained.

While this particular parametrization may provide some benefits, the lack of engagement with relevant prior work and other stochastic baselines significant limits the impact of this work and makes assessing its significance difficult.

This work would benefit from careful copyediting.

[1] Heess, N., Wayne, G., Silver, D., Lillicrap, T., Erez, T., & Tassa, Y. (2015). Learning continuous control policies by stochastic value gradients. In Advances in Neural Information Processing Systems (pp. 2944-2952).

[2] Mnih, V., Badia, A. P., Mirza, M., Graves, A., Lillicrap, T., Harley, T., ... & Kavukcuoglu, K. (2016, June). Asynchronous methods for deep reinforcement learning. In International Conference on Machine Learning (pp. 1928-1937).

[3] Schulman, J., Moritz, P., Levine, S., Jordan, M., & Abbeel, P. (2015). High-dimensional continuous control using generalized advantage estimation. arXiv preprint arXiv:1506.02438.

---

> ### Author Response · Authors · 2018-01-05
> **Response to Reviewer2**
>
>
> Thank you for your feedback and the additional references. We have now added and discussed your suggestions. We reply to your review in detail.
>
> >>> The wide use of stochastic policies in prior work makes the introductory explanation of the potential benefits for stochastic policies distracting, instead the focus should be on the particular choice and benefits of the particular stochastic parameterization chosen here and the choice of stochastic value gradient as a training method
>
> In addition to the suggested  references we have cleaned up and improved this section. Further we have added an experiment where we train without an exploration mechanism and can show, that the stochasticity of our method is enough to sufficiently explore the solution space in the beginning of training. MAPG without an explicit exploration mechanism still achieves a good performance. For some tasks DDPG however, is unable to learn a good policy without exploration (see also our reply to R1 regarding exploration).
> Further, we have emphasized the theoretical benefits of our method.
>
> >>>  The algorithm here can be seen as SVG(0) with a particular parametrization of the policy.
>
> We have carefully compared SVG(0) with our method but do not see much similarity between the two algorithms. Could you clarify your thoughts here? SVG(0) learns a variance for an action while we predict multiple actions that do not necessarily follow a given (Gaussian) distribution.
>
> >>> DDPG performance here is lower for several environments than the results reported in Henderson et al. 2017
>
> Thank you for the hint. We have made clear in the paper that this difference comes from the fact, that we train for 2,000 epochs for all environments and M-values for a fair and reproducible experimental setup.
>
> >>> The empirical comparison is also hampered by only comparing with DDPG
>
> Thank you! We have added A3C results to the Table 2.

---

> > ### Comment · AnonReviewer2 · 2018-01-12
> > **Separation of results**
> >
> > The relationship with SVG0, is that both are off-policy stochastic algorithms learned with the reparametrization trick. Currently the comparisons you have are with DDPG (deterministic, off-policy), A3C(stochastic, on-policy) and MAPG(stochastic, on-policy). So it is difficult to separate which gains are simply due to stochastic, off-policy learning and which might be due to the specific multi-modal distribution used.
> >
> > Overall, the paper reads significantly better now and does a better job of placing this work in the context of earlier results, but I believe it will be limited interest and still misses a key control.

---

### Official Review · AnonReviewer3 · 2017-11-27

**Rating:** 7
**Confidence:** 3

**Review:**

In this paper, the authors investigate a simple method for improving the performance of networks trained with DDPG: instead of outputting a single action, they output several actions (through distinct output layers), and choose one uniformly at random. The selected action is updated using deterministic policy gradient. The critic Q is updated with a Bellman backup where the the choice of the action is marginalized out. Authors show improved performance on a large number of standard continuous control environment (openAI gym and TORCS).

The paper is well written, and the idea seems to work perhaps surprisingly well. The authors do a good job of investigating the behavior of their algorithm (in particular the increase of standard deviation in states where multiple optimal actions exist).

Similar ideas (mixture of gaussians for action distribution in policy gradient setups, or multi-modal action distribution through the use of latent variables) are often difficult to make work - I am curious why this particular method works so well.
In particular, it would be interesting to investigate how the algorithm avoids collapsing all actions into the same one; as implied by section 3.2, in a state with multiple optimal actions, there is no difference in loss between having all actions be nearly identical (and optimal), and all actions being distinct optimal actions. Furthermore, as the loss does not encourage diversity, once two actions are set to be similar in a state, intuitively it would be hard for the actions to become distinct again. Imagine for instance the cart pole problem with M=2. If both action layers start with the same 'tendency' (towards clock-wise or counter clock-wise motion), it is likely that the same tendency would be reinforced for both, and the network with M=2 would end up having a similar behavior to a classical network with M=1.

Is this problem avoided by using a large value of M? It would be interesting to investigate the behavior of the algorithm in a toy environment (perhaps a simple 2d navigation  task with distinct 'paths' with same cost) where the number of distinct basins of optimality is know for various states, and investigate in more details how diversity is maintained (perhaps as a function of M).


Minor:
- typo rho -> $\rho$
- Given the paper fits comfortably within the page limit, it would have been worthwhile to give mathematical details to the Algorithm 1 box (even if they are easy to find in text or appropriate references)

---

> ### Author Response · Authors · 2018-01-05
> **Response to Reviewer3**
>
>
> Thank you for your comments! We are happy that you value the simplicity of our approach and like the paper. We will reply to your comments in detail.
>
> >>> Similar ideas (mixture of gaussians for action distribution in policy gradient setups, or multi-modal action distribution through the use of latent variables) are often difficult to make work - I am curious why this particular method works so well.
>
> We have experimented with Gaussian mixtures also in supervised learning tasks and found that they can suffer from numerical instabilities especially in higher dimensions due to the log() and exp() terms. Consider the scenario where exactly one action that is optimal. A mixture model would need to predict (close to) zero variance which can easily cause instabilities. Our method, however, does not change the original model at all in terms of loss computation and gradients. The sampling is virtually transparent to the computation inside the model.
> Another reason - that we found after comments from R1 - is that our method naturally explores the action space, as initially the actions are randomly distributed due to the initialization (see also our reply to R1 regarding exploration).
>
> >>> In particular, it would be interesting to investigate how the algorithm avoids collapsing all actions into the same one; as implied by section 3.2, in a state with multiple optimal actions, there is no difference in loss between having all actions be nearly identical (and optimal), and all actions being distinct optimal actions.
>
> This is an interesting observation. We will explain our intuition. In loss space optimal actions are minima in different locations. Since our network is randomly initialized all M actions are distributed over the action space in random locations (in our experiments it helps when actions are normalized for example to [-1,1]). During training with gradient descent, each action will move to a close (local) minimum which is unlikely the same for all of them if multiple of them exist. In theory you are correct that nothing explicitly prevents the model for learning one single optimal action for all M proposals.
>
> >>> Furthermore, as the loss does not encourage diversity, once two actions are set to be similar in a state, intuitively it would be hard for the actions to become distinct again
>
> Before writing the paper, we have extensively discussed adding a diversity term to the actions, that encourages actions to be distinct. However, this would act against learning an optimal policy, which would invalidate our theoretical guarantee that all M action proposals have the same expected performance. Diversity would be a trade-off for performance. One option would be to decay the diversity term over time, but since the initialization is already diverse, we do not expect this to have a big influence. Separating two identical actions can be done either with exploration noise or naturally in a non static, stochastic environment, where the same action in the same current state could receive different rewards.
>
> >>> It would be interesting to investigate the behavior of the algorithm in a toy environment (perhaps a simple 2d navigation  task with distinct 'paths' with same cost) where the number of distinct basins of optimality is know for various states, and investigate in more details how diversity is maintained (perhaps as a function of M).
>
> This is indeed an interesting experiment that we will investigate in the future. The exploration experiment that we have added to the paper shows the same property. During training without exploration, initially equally scoring actions are kept around making the agent explore more possibilities until it is able to find a good strategy. This means that not only the final policy benefits from multiple actions, but also the training is improved since the network does not need to decide for one out of several possible “paths” early.
>
> >>> typo rho -> $\rho$
>
> Thank you! We have fixed it.
>
> >>> Given the paper fits comfortably within the page limit, it would have been worthwhile to give mathematical details to the Algorithm 1 box (even if they are easy to find in text or appropriate references)
>
> We have extended Algorithm 1 with more details.

---

### Official Review · AnonReviewer1 · 2017-11-29
**Interesting idea but insufficient analysis.**

**Rating:** 4
**Confidence:** 4

**Review:**

This paper describes an approach to stochastic control using RL that extends DDPG with a stochastic policy.  A standard DDPG setup is extended such that the actor now produces M actions at each timestep.  Only one of the M actions will be executed in the environment using a uniform sampling.  The sampled action is the only that will receive a gradient update from the critic network. The authors demonstrate that such a stochastic policy performs better on average in a series of benchmark control tasks.

I find the general idea of the work compelling, but the particular approach is rather poor.  The fact that we are choosing the number of modes in the uniform distribution is a bit underwhelming (a more compelling architecture could have proposed a policy conditioned on gaussian noise for example, thus having better coverage of the distribution). I found the proposed apprach to be under-analyzed and the stochastic aspects of the policy are undervalued.   The main claim being argued in the paper is that the proposed stochastic policy has better final performance on average than a deterministic policy, but the only practical difference seems to be a slightly more structured approach to exploration.
However, very little attention is paid to trying different exploration methods with the deterministic policy (incidentally, Ornstein-Uhlenbeck process noise is not something I'm familiar with, a citation to the use of this noise for exploration as well as a more explicit explanation would have been appreciated).  One interpretation is that each of the M sub-policies follows a different mode of the Q-value distribution over the action space.  But is this indeed the case?  There is a brief analysis of this with cartpole, but a more complete look at how actions are clustered in the action space would make this paper much more compelling.  Even in higher-dimensional action spaces, you could look at a t-SNE projection or cluster analysis to try and see how many modes the agent is reasoning over.  Additionally, the baseline agent should have used additional exploration methods as these can quickly change the performance of the agent.

I also think that better learning is not the only redeeming aspect of a stochastic policy.  In the face of a non-stationary environment, a stochastic policy will likely be much more robust.  Additionally, it will have much better performance against adversarial environments.  Given the remaining available space in the paper it would have been interesting to provide more insight into the proposed methods gains in these areas.

---

> ### Author Response · Authors · 2018-01-05
> **Response to Reviewer1**
>
> Thank you for your comments! We are pleased to hear that you are intrigued by our approach. We have made changes to the paper (including additional experiments) based on your suggestions. We will reply to your comments in detail in the following.
>
> >>> choosing the number of modes in the uniform distribution is a bit underwhelming (a more compelling architecture could have proposed a policy conditioned on gaussian noise for example, thus having better coverage of the distribution)
>
> It is indeed possible to learn a parametrized distribution, for example by predicting a Gaussian mixture model (e.g. by adapting Mixture Density Networks, C. M. Bishop, 1994). However, we see two reasons why our simple approach is compelling. First, with our method we do not constrain the actions to follow a predefined distribution, thus it can in theory learn any optimal action sub-space. Second, usually parametrized distributions come with numerical difficulties in high dimensional output spaces. Our model is easy to train since it does not change the original model with additional numerically challenging computations (e.g. exp() in GMMs). (see also our reply to R3 on predicting distributions)
>
> >>> very little attention is paid to trying different exploration methods with the deterministic policy
>
> Thank you for this hint! We have added an experiment where we do not use any exploration during training (newly added Section 4.4). In the Pendulum environment, DDPG is not able to explore sufficiently, especially in the beginning of training, and shows poorer performance, while MAPG converges to a high performance. As you suspected, this shows that the stochastic nature of our method helps not only to learn a better policy but also results in better exploration during training. In the Cheetah environment we see that exploration helps late during training, indicating that is helps the M actions to become diverse.
>
> >>> Ornstein-Uhlenbeck process noise is not something I'm familiar with, a citation to the use of this noise for exploration as well as a more explicit explanation would have been appreciated
>
> Thank you! We have refined this section in the paper and added a reference. Further we analyze the performance of our method in the added Section 4.4 without exploration.
>
> >>> One interpretation is that each of the M sub-policies follows a different mode of the Q-value distribution over the action space.
>
> This is correct. We have improved Figure 3 to be more easily readable, where we visualize this intuition for the Pendulum task. The interesting aspect of MAPG is that it does not parametrize the distribution, thus it can in theory learn a point-wise approximation for any distribution.
>
> >>> a stochastic policy will likely be much more robust. Additionally, it will have much better performance against adversarial environments.
>
> We agree that this might be the case and are eager to try this!

---

### Author Response · Authors · 2018-01-05
**Added revision incorporating reviewer feedback**

We have updated the paper based on the feedback and comments of all reviewers. Here we list the major changes in the manuscript:

 - Changed abstract and introduction to reflect that MAPG is a general policy gradient algorithm.

 - Updated Algorithm 1 with equations and more details.

 - Added Section 4.4, exploring the effect of exploration noise onto the training together with Figure 4.

 - Improved Figure 3.

 - Added additional references (e.g. A3C).

 - Added A3C results to Table 2.

 - Revised introduction.

 - Several small corrections of typos.

---

### Decision · Program_Chairs · 2018-01-29
**ICLR 2018 Conference Acceptance Decision**

**Decision:**

Reject

**Comment:**

All of the reviewers agree that the paper presents strong experimental results on continuous control benchmarks. The reviewers raised concerns regarding the analysis of the behavior of the algorithm, the possible impact of the technique, and requested more references and comparison with related work. The paper has significantly improved since the initial submission, but still not able fully satisfactory to the reviewers, partly due to the large extent of the changes needed.